# Visible-Light-Induced Decarboxylation of Dioxazolones to Phosphinimidic Amides and Ureas

**DOI:** 10.3390/molecules27123648

**Published:** 2022-06-07

**Authors:** Jie Pan, Haocong Li, Kai Sun, Shi Tang, Bing Yu

**Affiliations:** 1Green Catalysis Center, College of Chemistry, Zhengzhou University, Zhengzhou 450001, China; 202012152012704@gs.zzu.edu.cn (J.P.); l348855465@gs.zzu.edu.cn (H.L.); 2College of Chemistry & Materials Engineering, Huaihua University, Huaihua 418008, China; 3College of Chemistry and Chemical Engineering, Jishou University, Jishou 416000, China; stang@jsu.edu.cn

**Keywords:** dioxazolones, visible light, decarboxylation, phosphinimidic amides, ureas

## Abstract

A visible-light-induced external catalyst-free decarboxylation of dioxazolones was realized for the bond formation of N=P and N–C bonds to access phosphinimidic amides and ureas. Various phosphinimidic amides and ureas (47 examples) were synthesized with high yields (up to 98%) by this practical strategy in the presence of the system’s ppm Fe.

## 1. Introduction

Nowadays, the development of clean, environmentally friendly, and efficient chemical processes has become one of the goals of sustainable chemistry [1,2,3,4]. Visible light, as a safe, abundant, and renewable natural energy source, has promoted many feasible and valuable transformations [5,6,7,8]. Photocatalytic strategies were widely recognized as an attractive “green synthesis pathway” in organic transformations, which are promising from the standpoint of an environmentally friendly and sustainable perspective [9,10,11,12,13,14,15,16,17,18]. Despite the simple operation and mild reaction conditions, a precious metal complex or a synthetically elaborate organic dye is usually required [19,20,21,22]. It is of great significance to develop cleaner and greener photochemical pathways in the external catalyst-free protocol [23,24,25,26,27].

Nitrene intermediates have attracted great interest from chemists, due to their high reactivity [28,29,30,31,32,33,34,35,36]. Nitrene-based transformations allow the direct installation of nitrogen-containing building blocks into molecular backbones to build structurally complex compounds [37,38,39,40]. In the past few decades, a series of nitrene precursors were reported, such as organic azides [41,42], iminoiodinanes [43,44], amide N-O compounds [45,46], dioxazolones [47,48,49,50], and so on. Among them, dioxazolones are highly attractive because of their high activity, stability, convenience, and high coordination ability [51,52,53]. Herein, we present a visible-light-induced strategy to build N=P and N–C bonds for the generation of phosphinimidic amides and ureas from the reaction of dioxazolones and triarylphosphines or secondary amines (Figure 1). The transformations are realized without any other catalyst or additive at room temperature. The ppm Fe in the reactants, confirmed by ICP-MS, might play an important role in this reaction.

## 2. Results and Discussion

### Optimization of the Reaction Conditions

We commenced our study with the model reaction between 3-phenyl-1,4,2-dioxazol-5-one (**1a**) and triphenylphosphine (**2a**) under visible light and N_2_ atmosphere. The results are shown in Table 1. Initially, the reaction was carried out by employing DCE as the solvent under irradiation of 10 W 430 nm blue LED at room temperature, and the desired product *N*-(triphenyl-λ^5^-phosphinylidene)benzamide (**3a**) could be detected in 11% yield (entry 1). Afterwards, the solvent effect on the yield was investigated (entries 2–8). Different solvents, such as 1,4-dioxane, CH_3_OH, acetone, CH_3_CN, DMF, THF, and CH_2_Cl_2_, were surveyed, and the reaction exhibited excellent reaction performance in CH_2_Cl_2_ to provide the target product in 81% yield (entry 8). Further examination of the wavelengths of LED and substrate ratios showed no more positive results (entries 9–14). Control reactions confirmed that nearly no amidation product **3a** was detected at room temperature in the absence of visible light (entry 15). Moreover, when the reaction was carried out in the air, only a trace amount of the product **3a** was detected (entry 16). Therefore, the optimized reaction conditions were illustrated as follows: **1a** (0.1 mmol); **2a** (0.1 mmol); and CH_2_Cl_2_ (1 mL) in a N_2_ atmosphere under the irradiation of 430 nm blue LED (10 W) for 24 h at room temperature (entry 8).

With the optimized conditions in hand, the scope of the organophosphorus compounds and dioxazolones **1** was investigated (Figure 2). To our delight, various 3-phenyl dioxazolones bearing different electron-donating groups (-CH_3_, -*^t^*Bu, and -OCH_3_) or electron-withdrawing groups (-CF_3_, -F, -Cl, and -CN) on the phenyl ring at different positions could react smoothly with **2a** to produce the desired products (**3a**–**3n**) in moderate to excellent yields (42–98%). Among these cases, a slight steric hindrance effect was observed, and para-substituted 3-phenyl dioxazolones (**3a**–**3h**, 63–98%) showed higher reaction reactivities than those of ortho-substituted 3-phenyl dioxazolones (**3m**–**3n**, 42–47%). Moreover, the desired products **3o** and **3p**, which contain the skeletons of thiophene and furan, could also be successfully obtained in 43% and 50% of the yields, respectively. Additionally, electron-poor and electron-rich triphenylphosphine derivatives were all applicable to this transformation to access the desired products (**3q**–**3v**) in 51–91% yields. In addition, the phosphorus ligand, 1.1′-binaphthyl-2.2′-diphenylphosphine (BINAP), was also a suitable substrate to react with **1a**, providing the corresponding product **3w** in 51% yield.

Reaction conditions: **1a** (0.2 mmol); **2a** (0.2 mmol) in solvent (2 mL) at room temperature for 24 h under the irradiation of 10 W 430 nm blue LED. Isolated yields were given.

Then, we expanded the photocatalytic decarboxylation reaction of dioxazolones to the synthesis of unsymmetrical urea compounds (Figure 3). To our delight, a wide range of 3-phenyl dioxazolones all reacted efficiently with diisopropylamine **4a** to furnish the corresponding aryl ureas (**5a**–**5m**) in moderate to excellent yields (37–98%). In these cases, 3-phenyl dioxazolones bearing electron-donating groups (-CH_3_, -*^t^*Bu, and -OCH_3_) showed a better reaction efficiency than those of 3-phenyl dioxazolones bearing electron-withdrawing groups (-CF_3_, -F, -Cl). Moreover, the broad scope of the commercially available secondary amines all reacted smoothly in this transformation, adding to the formation of desired ureas (**5n**–**5v**) in good to excellent yields (80–96%). In addition, the primary amine, such as aniline, was also a suitable substrate for reaction with **1a**, providing the corresponding product **5w** in 52% yield. However, cyclohexylamine (**4l**) and benzylamine (**4m**) were not suitable in this transformation to react with **1a** to access the corresponding products **5x** and **5y**. Compared with the previous report [32], our method effectively avoids the harsh conditions of high temperature, showing good sustainability.

To our satisfaction, this method is also suitable for the reaction between 3-(*p*-tolyl)-1,4,2-dioxazol-5-one **1b** and 1,3-diphenylpropane-1,3-dione **6** to give the corresponding amide product in 58% yield (Figure 4a), which was previously reported in the presence of additional FeCl_3_ catalyst [29]. To verify the practicability of this synthetic protocol, the gram-scale synthesis of **3a** was carried out (for details, see the Appendix A). When the reaction was performed at a 5 mmol scale, the desired product **3a** was isolated in 80% yield, indicating that this approach has a good practicability and application prospect (Figure 4b).

Furthermore, we also evaluated the sensitivity of the reaction of **1a** and **2a**. Compared with the standard conditions, the changes in concentration, temperature, oxygen level, water level, light intensity, and scale were measured. The yields were measured by ^31^P NMR and the yield deviation was calculated (for details, see the Appendix A). Among them, light intensity and oxygen levels are important parameters for the reaction. Moreover, this transformation is moderately sensitive to water. Other parameters, such as concentration and temperature, can be regarded as random errors, which have a negligible impact on reaction efficiency (Appendix A).

Next, we calculated the E-factor [54,55] and EcoScale scores [56,57] of the chemical process to evaluate the safety, economic, and ecological properties of the method. The results are summarized in Appendix A. As can be seen, the E-factor is extremely low at 0.38 and 0.82, respectively, and the EcoScale penalty is also low, at 21.5 and 15.5. Both parameters reflect the excellent green chemistry metrics of the protocol.

To understand the mechanism of this transformation, a set of control experiments were performed (Figure 5). The phosphorylation of 4-methylbenzamide **8** with triphenylphosphine **2a** was performed to determine whether the N=P bond was formed through the amide intermediate. However, 4-methyl-*N*-(triphenyl-*λ*^5^-phosphaneylidene) benzamide **3b** was not detected (Figure 5a). Moreover, intermolecular competition experiments of **1a** and **8** were conducted, and only product **3a** was obtained with 48% yield (Figure 5b). These results demonstrated that the phosphorylation of dioxazolones was not conducted through amide intermediates. Furthermore, various radical trapping experiments were conducted (Figure 5c). When (2,2,6,6-tetramethylpiperidine-1-yl)oxidanyl (TEMPO) was added to the model reaction under standard conditions, the reaction was significantly inhibited. The TEMPO-trapped acyl nitrene adducts were detected by high-resolution mass spectrometry (HRMS), with peaks at 277.1922 *m/z*. Subsequently, when another radical scavenger, 2,6-di-*tert*-butyl-4-methylphenol (BHT), was subjected under standard conditions, the reaction was also severely suppressed, indicating a radical process in the phosphorylation of dioxazolone with triphenylphosphine. Then, the radical trapping experiments of **1a** and **4a** were conducted (Figure 5c). The decreased yields of product **5a** indicated that the transformation also involved a radical process.

In 2021, Yu and Bao et al., disclosed that FeCl_3_ (15 mol%) was required for the imidization of phosphines with dioxazolones under visible light irradiation [29]. While in our case, the transformations worked very well without any other additives. Considering the contamination issues in coupling reactions [58], we reasoned that some iron contamination might be possible in the manufacture of the starting materials. Therefore, the model reaction mixture was analyzed with inductively coupled plasma mass spectrometry (ICP-MS). Consequently, it is found that the Fe content of the reactions for the preparation of phosphinimidic amide (**3a**) and urea (**5a**) is approximately 27 ppm and 3 ppm, respectively (for details, see the Appendix A). ICP-MS experiments were also performed on the starting materials of the model reactions (dioxazolone, PPh_3_, and amine), and the results showed that the iron contents of the dioxazolone, PPh_3_, and amine were 123 ppm, 420 ppm, and 0.9 ppm, respectively (for details, see the Appendix A). It is reasoned that iron contamination issues in commercial chemicals are unavoidable during the production and transportation processes. When additional iron catalyst FeCl_3_ (5 mol%) was added to the model reaction under standard conditions, the reaction time was shortened and the yield was increased. These results confirmed that this reaction could be facilitated by iron catalysis (for details, see the Appendix A). These results suggest that, although it is not a real transition-meta-free system, it is still a synthetically useful procedure for the synthesis of phosphinimidic amides and ureas, especially from an industrial chemistry standpoint.

Based on these control experiments and previous literature reports, a plausible reaction pathway is proposed in Figure 6. Initially, the N atom of dioxazolones **1** coordinates with the Fe center to form complex **B**, which is excited by visible light to generate the highly active iron-aminyl radical **C** with the release of CO_2_. Subsequently, radical **C** reacts with triphenylphosphine **2a** to form the complex **D**, followed by a reduction and elimination process to obtain product **3**. On the other hand, intermediate **C** underwent Curtius rearrangement to form intermediate **E**, which further reacts with secondary amines **4** to obtain product **5**.

## 3. Experimental Section

### 3.1. General Information

All nuclear magnetic resonance (NMR) spectra were recorded on a Bruker Avance 400 MHz in CDCl_3_ at room temperature (20 ± 3 °C), by using tetramethylsilane as the internal standard. High-resolution mass spectra (HRMS) were conducted on a 3000-mass spectrometer, using Waters Q-Tof MS/MS system with the ESI technique.

Photochemical reactions were carried out under visible light irradiation by a blue LED at 25 °C. The RLH-18 8-position Photo Reaction System manufactured by Beijing Roger Tech Ltd. was used in this system (Appendix A). Eight 10 W blue LEDs were equipped in this photochemical reactor. The wavelength for blue LED is 430 nm, peak width at half-height is 18.4 nm (Appendix A). The distance from the light source to the irradiation vessel was approximately 15 mm.

### 3.2. General Experimental Procedures for the Synthesis of (***3a–3w***)

In a 25 mL reaction tube, dioxazolones **1** (0.2 mmol, 1.0 equiv), organic phosphine substrate **2** (0.2 mmol, 1.0 equiv) in 1 mL CH_2_Cl_2_ were allowed to stir with irradiation of 10 W blue LED under N_2_ atmosphere at room temperature for 24 h. After the reaction, the solvent was evaporated under vacuum, and the residue was purified by column chromatography on silica gel to afford the desired products **3a**–**3w**.

*N*-(triphenyl-*λ*^5^-phosphanylidene)benzamide (**3a**):

White solid (59.7 mg, 78%). ^1^H NMR (400 MHz, Chloroform-*d*) δ 8.45–8.38 (m, 2H), 7.94–7.85 (m, 6H), 7.62–7.55 (m, 3H), 7.54–7.41 (m, 9H); ^13^C NMR (101 MHz, Chloroform-*d*) δ 176.3 (d, *J_C-P_* = 8.0 Hz), 138.7 (d, *J_C-P_* = 20.6 Hz), 133.2 (d, *J_C-P_* = 10.0 Hz), 132.3 (d, *J_C-P_* = 2.9 Hz), 130.7, 129.6 (d, *J_C-P_* = 2.6 Hz), 128.7 (d, *J_C-P_* = 12.4 Hz), 128.4 (d, *J_C-P_* = 99.7 Hz), 127.7; ^31^P NMR (162 MHz, Chloroform-*d*) δ 20.71;

4-methyl-*N*-(triphenyl-*λ*^5^-phosphanylidene)benzamide (**3b**):

White solid (69.0 mg, 87%). ^1^H NMR (400 MHz, Chloroform-*d*) δ 8.30 (d, *J* = 8.2 Hz, 2H), 7.92–7.85 (m, 6H), 7.62–7.55 (m, 3H), 7.53–7.47 (m, 6H), 7.24 (d, *J* = 7.9 Hz, 2H), 2.42 (s, 3H); ^13^C NMR (101 MHz, Chloroform-*d*) δ 176.5 (d, *J_C-P_* = 8.1 Hz), 140.9, 135.9 (d, *J_C-P_* = 20.5 Hz), 133.2 (d, *J_C-P_* = 9.7 Hz), 132.2 (d, *J_C-P_* = 2.9 Hz), 129.6 (d, *J_C-P_* = 2.6 Hz), 128.7 (d, *J_C-P_* = 12.1 Hz), 128.5 (d, *J_C-P_* = 99.6 Hz), 128.4, 21.6; ^31^P NMR (162 MHz, Chloroform-*d*) δ 20.47;

4-(tert-butyl)-*N*-(triphenyl-*λ*^5^-phosphanylidene)benzamide (**3c**):

White solid (55.0 mg, 63%). ^1^H NMR (400 MHz, Chloroform-*d*) δ 8.33 (d, *J* = 8.5 Hz, 2H), 7.92–7.84 (m, 6H), 7.61–7.55 (m, 3H), 7.53–7.45 (m, 8H), 1.38 (s, 9H); ^13^C NMR (101 MHz, Chloroform-*d*) δ 176.4 (d, *J_C-P_* = 8.1 Hz), 153.9, 136.0 (d, *J_C-P_* = 20.5 Hz), 133.2 (d, *J_C-P_* = 10.1 Hz), 132.2 (d, *J_C-P_* = 2.9 Hz), 129.4 (d, *J_C-P_* = 2.6 Hz), 128.7 (d, *J_C-P_* = 12.4 Hz), 128.5 (d, *J_C-P_* = 99.3 Hz), 124.6, 34.9, 31.4; ^31^P NMR (162 MHz, Chloroform-*d*) δ 20.24;

4-methoxy-*N*-(triphenyl-*λ*^5^-phosphanylidene)benzamide (**3d**):

White solid (79 mg, 96%). ^1^H NMR (400 MHz, Chloroform-*d*) δ 8.35 (d, *J* = 8.8 Hz, 2H), 7.91–7.84 (m, 6H), 7.60–7.54 (m, 3H), 7.52–7.46 (m, 6H), 6.94 (d, *J* = 8.8 Hz, 2H), 3.85 (s, 3H); ^13^C NMR (101 MHz, Chloroform-*d*) δ 176.1 (d, *J_C-P_* = 7.8 Hz), 161.8, 133.2 (d, *J_C-P_* = 9.9 Hz), 132.2 (d, *J_C-P_* = 2.9 Hz), 131.5, 131.4 (d, *J_C-P_* = 2.5 Hz), 128.7 (d, *J_C-P_* = 12.2 Hz), 128.5 (d, *J_C-P_* = 99.6 Hz), 112.8, 55.3; ^31^P NMR (162 MHz, Chloroform-*d*) δ 20.30;

4-(trifluoromethyl)-*N*-(triphenyl-*λ*^5^-phosphanylidene)benzamide (**3e**):

White solid (87.6 mg, 94%). ^1^H NMR (400 MHz, Chloroform-*d*) δ 8.49 (d, *J* = 8.1 Hz, 2H), 7.91–7.84 (m, 6H), 7.69 (d, *J* = 8.1 Hz, 2H), 7.64–7.57 (m, 3H), 7.56–7.49 (m, 6H); ^13^C NMR (101 MHz, Chloroform-*d*) δ 174.8 (d, *J_C-P_* = 7.7 Hz), 142.0 (d, *J_C-P_* = 21.0 Hz), 133.2 (d, *J_C-P_* = 10.3 Hz), 132.5 (d, *J_C-P_* = 2.9 Hz), 132.2 (q, *J_C-F_* = 32.0 Hz), 129.8 (d, *J_C-P_* = 2.5 Hz), 128.8 (d, *J_C-P_* = 12.4 Hz), 127.9 (d, *J_C-P_* = 99.7 Hz), 124.7 (q, *J_C-F_* = 3.8 Hz), 124.3 (q, *J_C-F_* = 272.4 Hz); ^31^P NMR (162 MHz, Chloroform-*d*) δ 21.40; ^19^F NMR (376 MHz, Chloroform-*d*) δ −62.47;

4-fluoro-*N*-(triphenyl-*λ*^5^-phosphanylidene)benzamide (**3f**):

White solid (57.5 mg, 73%). ^1^H NMR (400 MHz, Chloroform-*d*) δ 8.38 (dd, *J* = 8.7, 5.9 Hz, 2H), 7.90–7.82 (m, 6H), 7.61–7.56 (m, 3H), 7.55–7.46 (m, 6H), 7.08 (t, *J* = 8.8 Hz, 2H); ^13^C NMR (101 MHz, Chloroform-*d*) δ 175.3 (d, *J_C-P_* = 8.0 Hz), 164.7 (d, *J_C-F_* = 249.4 Hz), 134.9 (d, *J_C-P_* = 20.1 Hz), 133.2 (d, *J_C-P_* = 10.1 Hz), 132.3 (d, *J_C-P_* = 2.9 Hz), 131.8 (dd, *J_C-F_* = 8.9, *J_C-P_* = 2.3 Hz), 128.7 (d, *J_C-P_* = 12.3 Hz), 128.2 (d, *J_C-P_* = 99.7 Hz), 114.4 (d, *J_C-F_* = 21.4 Hz); ^31^P NMR (162 MHz, Chloroform-*d*) δ 20.87; ^19^F NMR (376 MHz, Chloroform-*d*) δ −110.66;

4-chloro-*N*-(triphenyl-*λ*^5^-phosphanylidene)benzamide (**3g**):

White solid (81.8 mg, 98%). ^1^H NMR (400 MHz, Chloroform-*d*) δ 8.33 (d, *J* = 8.5 Hz, 2H), 7.91–7.82 (m, 6H), 7.62–7.55 (m, 3H), 7.54–7.48 (m, 6H), 7.39 (d, *J* = 8.5 Hz, 2H); ^13^C NMR (101 MHz, Chloroform-*d*) δ 175.2 (d, *J_C-P_* = 7.9 Hz), 137.2 (d, *J_C-P_* = 21.2 Hz), 136.8, 133.2 (d, *J_C-P_* = 9.7 Hz), 132.4 (d, *J_C-P_* = 2.9 Hz), 131.1 (d, *J_C-P_* = 2.3 Hz), 128.8 (d, *J_C-P_* = 12.4 Hz), 128.1 (d, *J_C-P_* = 99.7 Hz), 127.8; ^31^P NMR (162 MHz, Chloroform-*d*) δ 21.05;

4-cyano-*N*-(triphenyl-*λ*^5^-phosphaneylidene)benzamide (**3h**):

White solid (57.6 mg, 71%), mp 185.5–187.1 °C. ^1^H NMR (400 MHz, Chloroform-*d*) δ 8.44 (d, *J* = 8.1 Hz, 2H), 7.89–7.80 (m, 6H), 7.70 (d, *J* = 8.3 Hz, 2H), 7.64–7.57 (m, 3H), 7.56–7.49 (m, 6H); ^13^C NMR (101 MHz, Chloroform-*d*) δ 174.2 (d, *J_C-P_* = 8.1 Hz), 142.8 (d, *J_C-P_* = 21.3 Hz), 133.1 (d, *J_C-P_* = 9.7 Hz), 132.6 (d, *J_C-P_* = 3.0 Hz), 131.6, 130.0 (d, *J_C-P_* = 2.3 Hz), 128.9 (d, *J_C-P_* = 12.4 Hz), 127.7 (d, *J_C-P_* = 99.8 Hz), 119.1, 113.8; ^31^P NMR (162 MHz, Chloroform-*d*) δ 21.79. HRMS (ESI-TOF) *m/z*: [M + H]^+^ calcd for C_26_H_20_N_2_OP, 407.1308; found, 407.1309;

3-methoxy-*N*-(triphenyl-*λ*^5^-phosphanylidene)benzamide (**3i**):

White solid (62.4 mg, 76%). ^1^H NMR (400 MHz, Chloroform-*d*) δ 8.05 (d, *J* = 7.6 Hz, 1H), 7.93–7.83 (m, 7H), 7.61–7.55 (m, 3H), 7.54–7.47 (m, 6H), 7.35 (t, *J* = 7.9 Hz, 1H), 7.06–7.01 (m, 1H), 3.88 (s, 3H); ^13^C NMR (101 MHz, Chloroform-*d*) δ 176.1 (d, *J_C-P_* = 8.1 Hz), 159.3, 140.2 (d, *J_C-P_* = 20.5 Hz), 133.2 (d, *J_C-P_* = 9.7 Hz), 132.3 (d, *J_C-P_* = 2.9 Hz), 128.7 (d, *J_C-P_* = 12.5 Hz), 128.3 (d, *J_C-P_* = 99.2 Hz), 122.3 (d, *J_C-P_* = 2.6 Hz), 117.4, 113.8 (d, *J_C-P_* = 2.9 Hz), 55.4; ^31^P NMR (162 MHz, Chloroform-*d*) δ 20.70;

3-(trifluoromethyl)-*N*-(triphenyl-*λ*^5^-phosphanylidene)benzamide (**3j**):

White solid (65.1 mg, 73%). ^1^H NMR (400 MHz, Chloroform-*d*) δ 8.67 (s, 1H), 8.57 (d, *J* = 7.7 Hz, 1H), 7.92–7.84 (m, 6H), 7.72 (d, *J* = 8.1 Hz, 1H), 7.64–7.58 (m, 3H), 7.57–7.50 (m, 7H); ^13^C NMR (101 MHz, Chloroform-*d*) δ 174.7 (d, *J_C-P_* = 7.7 Hz), 139.5 (d, *J_C-P_* = 21.2 Hz), 133.2 (d, *J_C-P_* = 10.2 Hz), 132.8, 132.4 (d, *J_C-P_* = 3.0 Hz), 130.1 (q, *J_C-F_* = 32.2 Hz), 128.8 (d, *J_C-P_* = 12.4 Hz), 128.2, 128.0 (d, *J_C-P_* = 99.8 Hz), 127.1 (q, *J_C-F_* = 3.8 Hz), 126.5 (q, *J_C-F_* = 3.5 Hz), 123.0 (q, *J_C-F_* = 272.2 Hz); ^31^P NMR (162 MHz, Chloroform-*d*) δ 21.62; ^19^F NMR (376 MHz, Chloroform-*d*) δ −62.33;

3-fluoro-*N*-(triphenyl-*λ*^5^-phosphanylidene)benzamide (**3k**):

White solid (48.7 mg, 61%), mp 154.6–156.5 °C. ^1^H NMR (400 MHz, Chloroform-*d*) δ 8.14 (d, *J* = 7.7 Hz, 1H), 8.10–8.05 (m, 1H), 7.91–7.82 (m, 6H), 7.63–7.56 (m, 3H), 7.55–7.48 (m, 6H), 7.42–7.34 (m, 1H), 7.19–7.13 (m, 1H); ^13^C NMR (101 MHz, Chloroform-*d*) δ 175.0 (d, *J_C-P_* = 7.4 Hz), 162.6 (d, *J_C-F_* = 244.8 Hz), 141.2 (d, *J_C-F_* = 21.3 Hz), 133.2 (d, *J_C-P_* = 10.0 Hz), 132.4 (d, *J_C-F_* = 2.9 Hz), 129.1 (d, *J_C-P_* = 7.7 Hz), 128.8 (d, *J_C-P_* = 12.4 Hz), 128.1 (d, *J_C-P_* = 99.8 Hz), 125.1 (d, *J_C-P_* = 2.8 Hz), 117.5 (d, *J_C-P_* = 21.6 Hz), 116.4 (dd, *J_C-F_* = 22.2, *J_C-P_* = 2.6 Hz); ^31^P NMR (162 MHz, Chloroform-*d*) δ 21.13; ^19^F NMR (376 MHz, Chloroform-*d*) δ −114.38. HRMS (ESI-TOF) *m/z*: [M + H]^+^ calcd for C_25_H_20_FNOP, 400.1261; found, 400.1261;

3-chloro-*N*-(triphenyl-*λ*^5^-phosphanylidene)benzamide (**3l**):

Colorless liquid (56.7 mg, 64%). ^1^H NMR (400 MHz, Chloroform-*d*) δ 8.40–8.36 (m, 1H), 8.23 (d, *J* = 7.8 Hz, 1H), 7.90–7.82 (m, 6H), 7.63–7.57 (m, 3H), 7.55–7.48 (m, 6H), 7.46–7.41 (m, 1H), 7.37–7.32 (m, 1H); ^13^C NMR (101 MHz, Chloroform-*d*) δ 174.9 (d, *J_C-P_* = 8.0 Hz), 140.6 (d, *J_C-P_* = 21.2 Hz), 133.7, 133.2 (d, *J_C-P_* = 10.2 Hz), 132.4 (d, *J_C-P_* = 2.6 Hz), 130.6, 129.8 (d, *J_C-P_* = 2.7 Hz), 129.0, 128.8 (d, *J_C-P_* = 12.0 Hz), 128.0 (d, *J_C-P_* = 99.8 Hz), 127.6 (d, *J_C-P_* = 2.2 Hz); ^31^P NMR (162 MHz, Chloroform-*d*) δ 21.40. HRMS (ESI-TOF) *m/z*: [M + H]^+^ calcd for C_25_H_20_ClNOP, 416.0966; found, 416.0963;

2-fluoro-*N*-(triphenyl-*λ*^5^-phosphanylidene)benzamide (**3m**):

White solid (37.5 mg, 47%). ^1^H NMR (400 MHz, Chloroform-*d*) δ 8.22–8.16 (m, 1H), 7.93–7.84 (m, 6H), 7.61–7.55 (m, 3H), 7.53–7.47 (m, 6H), 7.41–7.34 (m, 1H), 7.18–7.06 (m, 2H); ^13^C NMR (101 MHz, Chloroform-*d*) δ 174.1 (dd, *J_C-P_ =* 7.9 Hz*, J_C-F_ =* 3.6 Hz), 161.9 (d, *J_C-F_* = 255.2 Hz), 133.2 (d, *J_C-P_* = 10.1 Hz), 132.4 (dd, *J_C-P_ =* 2.3 Hz, *J_C-F_ =* 2.2 Hz), 132.3 (d, *J_C-P_* = 2.9 Hz), 131.7 (d, *J_C-F_* = 8.8 Hz), 128.7 (d, *J_C-P_* = 12.4 Hz), 128.0 (d, *J_C-P_* = 99.3 Hz), 127.5 (dd, *J_C-P_ =* 21.4 Hz*, J_C-F_ =* 9.9 Hz), 123.3 (d, *J_C-F_* = 3.8 Hz), 116.5 (d, *J_C-F_* = 23.4 Hz); ^31^P NMR (162 MHz, Chloroform-*d*) δ 20.39; ^19^F NMR (376 MHz, Chloroform-*d*) δ −111.60;

2-chloro-*N*-(triphenyl-*λ*^5^-phosphanylidene)benzamide (**3n**):

White solid (35.0 mg, 42%), mp 196.6–197.8 °C. ^1^H NMR (400 MHz, Chloroform-*d*) δ 8.43–8.36 (m, 2H), 7.92–7.84 (m, 6H), 7.62–7.56 (m, 3H), 7.54–7.48 (m, 6H), 7.46–7.41 (m, 2H); ^13^C NMR (101 MHz, Chloroform-*d*) δ 176.4 (d, *J_C-P_* = 8.6 Hz), 138.6 (d, *J_C-P_* = 20.6 Hz), 133.2 (d, *J_C-P_* = 9.7 Hz), 132.3 (d, *J_C-P_* = 2.9 Hz), 130.7, 129.6 (d, *J_C-P_* = 2.3 Hz), 128.7 (d, *J_C-P_* = 12.3 Hz), 128.4 (d, *J_C-P_* = 99.5 Hz), 127.7; ^31^P NMR (162 MHz, Chloroform-*d*) δ 20.72. HRMS (ESI-TOF) *m/z*: [M + H]^+^ calcd for C_25_H_20_ClNOP, 416.0966; found, 416.0969;

*N*-(triphenyl-*λ*^5^-phosphanylidene)thiophene-2-carboxamide (**3o**):

Brown solid (33.3 mg, 43%). ^1^H NMR (400 MHz, Chloroform-*d*) δ 7.89–7.82 (m, 6H), 7.80 (dd, *J* = 3.6, 1.2 Hz, 1H), 7.62–7.56 (m, 3H), 7.53–7.47 (m, 6H), 7.40 (dd, *J* = 5.0, 1.2 Hz, 1H), 7.10–7.01 (m, 1H); ^13^C NMR (101 MHz, Chloroform-*d*) δ 171.1 (d, *J_C-P_* = 7.0 Hz), 145.2 (d, *J_C-P_* = 24.3 Hz), 133.2 (d, *J_C-P_* = 9.7 Hz), 132.3 (d, *J_C-P_* = 2.9 Hz), 130.2 (d, *J_C-P_* = 2.8 Hz), 129.6, 128.7 (d, *J_C-P_* = 12.4 Hz), 128.1 (d, *J_C-P_* = 99.7 Hz), 127.3; ^31^P NMR (162 MHz, Chloroform-*d*) δ 19.47;

*N*-(triphenyl-*λ*^5^-phosphanylidene)furan-2-carboxamide (**3p**):

White solid (37.2 mg, 50%). ^1^H NMR (400 MHz, Chloroform-*d*) δ 7.87–7.79 (m, 6H), 7.59–7.53 (m, 3H), 7.51–7.45 (m, 7H), 7.17 (d, *J* = 3.3 Hz, 1H), 6.44 (dd, *J* = 3.2, 1.7 Hz, 1H); ^13^C NMR (101 MHz, Chloroform-*d*) δ 168.0 (d, *J_C-P_* = 6.7 Hz), 152.8 (d, *J_C-P_* = 26.0 Hz), 144.0, 133.2 (d, *J_C-P_* = 10.1 Hz), 132.4, 128.7 (d, *J_C-P_* = 12.4 Hz), 127.9 (d, *J_C-P_* = 100.2 Hz), 114.3, 111.3; ^31^P NMR (162 MHz, Chloroform-*d*) δ 21.86;

*N*-(tri-*p*-tolyl-*λ*^5^-phosphanylidene)benzamide (**3q**):

White solid (77.1 mg, 91%). ^1^H NMR (400 MHz, Chloroform-*d*) δ 8.42 (d, *J* = 7.8 Hz, 2H), 7.79 (dd, *J* = 12.2, 7.7 Hz, 6H), 7.48–7.41 (m, 3H), 7.32 (dd, *J* = 8.3, 2.8 Hz, 6H), 2.43 (s, 9H); ^13^C NMR (101 MHz, Chloroform-*d*) δ 176.2 (d, *J_C-P_* = 8.0 Hz), 142.7 (d, *J_C-P_* = 2.9 Hz), 138.9 (d, *J_C-P_* = 20.8 Hz), 133.2 (d, *J_C-P_* = 10.3 Hz), 130.6, 129.6 (d, *J_C-P_* = 2.3 Hz), 129.4 (d, *J_C-P_* = 12.9 Hz), 127.6, 125.4 (d, *J_C-P_* = 102.0 Hz), 21.7; ^31^P NMR (162 MHz, Chloroform-*d*) δ 20.63;

*N*-(tri-*p*-methoxyphenyl-*λ*^5^-phosphanylidene)benzamide (**3r**):

White solid (56.7 mg, 60%). ^1^H NMR (400 MHz, Chloroform-*d*) δ 8.40–8.34 (m, 2H), 7.78 (dd, *J* = 11.7, 8.8 Hz, 6H), 7.46–7.38 (m, 3H), 7.00 (dd, *J* = 8.9, 2.3 Hz, 6H), 3.85 (s, 9H); ^13^C NMR (101 MHz, Chloroform-*d*) δ 176.1 (d, *J_C-P_* = 7.4 Hz), 162.6 (d, *J_C-P_* = 2.9 Hz), 139.0 (d, *J_C-P_* = 20.4 Hz), 135.0 (d, *J_C-P_* = 11.1 Hz), 130.5, 129.5 (d, *J_C-P_* = 2.3 Hz), 127.6, 119.9 (d, *J_C-P_* = 106.5 Hz), 114.3 (d, *J_C-P_* = 13.3 Hz), 55.4; ^31^P NMR (162 MHz, Chloroform-*d*) δ 19.68;

*N*-(tris(4-fluorophenyl)-*λ*^5^-phosphanylidene)benzamide (**3s**):

White solid (44.2 mg, 51%). ^1^H NMR (400 MHz, Chloroform-*d*) δ 8.34–8.29 (m, 2H), 7.90–7.81 (m, 6H), 7.50–7.41 (m, 3H), 7.26–7.19 (m, 6H); ^13^C NMR (101 MHz, Chloroform-*d*) δ 176.5 (d, *J_C-P_* = 8.1 Hz), 165.4 (dd, *J_C-F_* = 255.1 Hz, *J_C-P_* = 3.2 Hz), 138.1 (d, *J_C-P_* = 20.6 Hz), 135.6 (dd, *J_C-F_* = 11.7 Hz, *J_C-P_* = 8.8 Hz), 131.0, 129.5 (d, *J_C-P_* = 2.7 Hz), 127.8, 123.9 (dd, *J_C-P_* = 103.9 Hz, *J_C-F_* = 3.3 Hz), 116.4 (dd, *J_C-F_* = 21.6 Hz, *J_C-P_* = 13.6 Hz); ^31^P NMR (162 MHz, Chloroform-*d*) δ 18.98; ^19^F NMR (376 MHz, Chloroform-*d*) δ −105.30;

*N*-(tris(4-chlorophenyl)-*λ*^5^-phosphanylidene)benzamide (**3t**):

White solid (80.5 mg, 83%). ^1^H NMR (400 MHz, Chloroform-*d*) δ 8.36–8.30 (m, 2H), 7.79 (dd, *J* = 12.0, 8.4 Hz, 6H), 7.54–7.47 (m, 7H), 7.43 (dd, *J* = 8.1, 6.2 Hz, 2H); ^13^C NMR (101 MHz, Chloroform-*d*) δ 176.7 (d, *J_C-P_* = 8.0 Hz), 139.5 (d, *J_C-P_* = 3.6 Hz), 137.9 (d, *J_C-P_* = 20.5 Hz), 134.4 (d, *J_C-P_* = 11.0 Hz), 131.1, 129.5 (d, *J_C-P_* = 2.5 Hz), 129.4 (d, *J_C-P_* = 12.8 Hz), 127.8, 126.2 (d, *J_C-P_* = 102.0 Hz); ^31^P NMR (162 MHz, Chloroform-*d*) δ 19.49;

*N*-(tris(3-methoxyphenyl)-*λ*^5^-phosphaneylidene)benzamide (**3u**):

White solid (48.5 mg, 52%), mp 146.3–147.7 °C. ^1^H NMR (400 MHz, Chloroform-*d*) δ 8.41–8.36 (m, 2H), 7.52–7.47 (m, 3H), 7.46–7.34 (m, 9H), 7.12–7.07 (m, 3H), 3.79 (s, 9H); ^13^C NMR (101 MHz, Chloroform-*d*) δ 176.2 (d, *J_C-P_* = 7.9 Hz), 159.6 (d, *J_C-P_* = 15.4 Hz), 138.7 (d, *J_C-P_* = 20.6 Hz), 130.7, 129.9 (d, *J_C-P_* = 14.6 Hz), 129.6 (d, *J_C-P_* = 99.1 Hz), 129.5 (d, *J_C-P_* = 2.4 Hz), 127.7, 125.4 (d, *J_C-P_* = 9.7 Hz), 118.4 (d, *J_C-P_* = 11.0 Hz), 118.1 (d, *J_C-P_* = 2.9 Hz), 55.4; ^31^P NMR (162 MHz, Chloroform-*d*) δ 21.38. HRMS (ESI-TOF) *m/z*: [M + H]^+^ calcd for C_28_H_27_NO_4_P, 472.1672; found, 472.1677;

*N*-(diphenyl(*p*-tolyl)-*λ*^5^-phosphanylidene)benzamide (**3v**):

White solid (71.1 mg, 90%). ^1^H NMR (400 MHz, Chloroform-*d*) δ 8.45–8.37 (m, 2H), 7.93–7.85 (m, 4H), 7.81–7.74 (m, 2H), 7.61–7.55 (m, 2H), 7.53–7.42 (m, 7H), 7.32 (dd, *J* = 8.3, 2.9 Hz, 2H), 2.43 (s, 3H); ^13^C NMR (101 MHz, Chloroform-*d*) δ 176.3 (d, *J_C-P_* = 8.0 Hz), 142.9 (d, *J_C-P_* = 2.9 Hz), 138.7 (d, *J_C-P_* = 20.6 Hz), 133.3 (d, *J_C-P_* = 10.5 Hz), 133.2 (d, *J_C-P_* = 9.5 Hz), 132.2 (d, *J_C-P_* = 3.0 Hz), 130.7, 129.6, 129.5 (d, *J_C-P_* = 10.0 Hz), 128.7 (d, *J_C-P_* = 12.3 Hz), 128.6 (d, *J_C-P_* = 99.8 Hz), 127.7, 124.8 (d, *J_C-P_* = 101.3 Hz), 21.7; ^31^P NMR (162 MHz, Chloroform-*d*) δ 20.67;

*N,N*’-([1,1′-binaphthalene]-2,2′-diylbis(diphenyl-*λ*^5^-phosphaneylylidene))dibenzamide (**3w**):

White solid (87.7 mg, 51%), mp 184.5–185.8 °C. ^1^H NMR (400 MHz, Chloroform-*d*) δ 8.18 (d, *J* = 7.1 Hz, 4H), 7.78–7.69 (m, 6H), 7.57–7.46 (m, 7H), 7.44–7.29 (m, 13H), 7.28–7.21 (m, 3H), 7.20–7.07 (m, 7H), 6.41 (dd, *J* = 8.3, 5.0 Hz, 2H); ^13^C NMR (101 MHz, Chloroform-*d*) δ 175.9 (d, *J_C-P_* = 8.0 Hz), 159.6 (d, *J_C-P_* = 2.0 Hz), 138.8 (d, *J_C-P_* = 21.2 Hz), 135.1 (d, *J_C-P_* = 7.1 Hz), 134.1 (d, *J_C-P_* = 2.6 Hz), 133.3 (d, *J_C-P_* = 10.5 Hz), 132.9 (d, *J_C-P_* = 10.3 Hz), 132.1 (d, *J_C-P_* = 3.0 Hz), 131.6 (d, *J_C-P_* = 2.4 Hz), 130.4, 129.4 (d, *J_C-P_* = 2.7 Hz), 128.5 (d, *J_C-P_* = 12.5 Hz), 128.2 (d, *J_C-P_* = 12.6 Hz), 127.8 (d, *J_C-P_* = 104.7 Hz), 127.50, 127.45, 123.9 (d, *J_C-P_* = 11.4 Hz), 121.0 (d, *J_C-P_* = 6.8 Hz), 119.2 (d, *J_C-P_* = 100.4 Hz); ^31^P NMR (162 MHz, Chloroform-*d*) δ 19.95. HRMS (ESI-TOF) *m/z*: [M + H]^+^ calcd for C_58_H_43_N_2_O_2_P_2_, 861.2794; found, 861.2791.

### 3.3. General Experimental Procedures for the Synthesis of (***5a***–***5w***)

In a 25 mL reaction tube, dioxazolone **1** (0.2 mmol, 1.0 equiv.), and amine **4** (0.4 mmol, 2.0 equiv.) in 1 mL CH_3_OH were allowed to stir with irradiation of 10 W blue LED at room temperature for 5 h. After the reaction, the solvent was evaporated under vacuum, and the residue was purified by column chromatography on silica gel to afford the desired products **5a**–**5w**.

1,1-diisopropyl-3-phenylurea (**5a**):

White solid (40.1 mg, 91%). ^1^H NMR (400 MHz, Chloroform-*d*) δ 7.39 (dd, *J* = 8.4, 1.3 Hz, 2H), 7.31–7.26 (m, 2H), 7.06–6.99 (m, 1H), 6.26 (s, 1H), 4.04–3.95 (m, 2H), 1.34 (d, *J* = 7.0 Hz, 12H); ^13^C NMR (101 MHz, Chloroform-*d*) δ 154.6, 139.4, 128.8, 122.6, 119.7, 45.5, 21.5;

1,1-diisopropyl-3-(*p*-tolyl)urea (**5b**):

White solid (44.9 mg, 96%). ^1^H NMR (400 MHz, Chloroform-*d*) δ 7.27 (d, *J* = 8.3 Hz, 2H), 7.09 (d, *J* = 8.4 Hz, 2H), 6.18 (s, 1H), 4.04–3.94 (m, 2H), 2.30 (s, 3H), 1.33 (d, *J* = 6.9 Hz, 12H); ^13^C NMR (101 MHz, Chloroform-*d*) δ 154.8, 136.8, 132.1, 129.3, 119.9, 45.4, 21.5, 20.7;

3-(4-(*tert*-butyl)phenyl)-1,1-diisopropylurea (**5c**):

White solid (49.1 mg, 89%), mp 115.1–116.7 °C. ^1^H NMR (400 MHz, Chloroform-*d*) δ 7.33–7.29 (m, 4H), 6.17 (s, 1H), 4.05–3.96 (m, 2H), 1.34 (d, *J* = 6.9 Hz, 12H); 1.31 (s, 9H); ^13^C NMR (101 MHz, Chloroform-*d*) δ 154.9, 145.6, 136.7, 125.7, 119.7, 45.4, 34.2, 31.4, 21.6. HRMS (ESI-TOF) *m/z*: [M + H]^+^ calcd for C_17_H_29_N_2_O, 277.2274; found, 277.2284;

1,1-diisopropyl-3-(4-methoxyphenyl)urea (**5d**):

White solid (49.0 mg, 98%). ^1^H NMR (400 MHz, Chloroform-*d*) δ 7.27 (d, *J* = 8.9 Hz, 2H), 6.83 (d, *J* = 9.0 Hz, 2H), 6.13 (s, 1H), 4.01–3.92 (m, 2H), 3.77 (s, 3H), 1.32 (d, *J* = 6.9 Hz, 12H); ^13^C NMR (101 MHz, Chloroform-*d*) δ 155.5, 155.1, 132.5, 122.0, 114.1, 55.5, 45.4, 21.5;

1,1-diisopropyl-3-(4-(trifluoromethyl)phenyl)urea (**5e**):

White solid (44.2 mg, 79%), mp 151.9–153.1 °C. ^1^H NMR (400 MHz, Chloroform-*d*) δ 7.57–7.46 (m, 4H), 6.43 (s, 1H), 4.07–3.92 (m, 2H), 1.35 (d, *J* = 6.9 Hz, 12H); ^13^C NMR (101 MHz, Chloroform-*d*) δ 154.0, 142.6, 126.1 (q, *J_C-F_* = 3.8 Hz), 124.4 (q, *J_C-F_* = 271.1 Hz), 124.2 (q, *J_C-F_* = 32.7 Hz), 118.9, 45.7, 21.5; ^19^F NMR (376 MHz, Chloroform-*d*) δ −61.81; HRMS (ESI-TOF) *m/z*: [M + H]^+^ calcd for C_14_H_20_F_3_N_2_O, 289.1522; found, 289.1534;

3-(4-fluorophenyl)-1,1-diisopropylurea (**5f**):

White solid (36.7 mg, 77%), mp 134.5–135.7 °C. ^1^H NMR (400 MHz, Chloroform-*d*) δ 7.35–7.27 (m, 2H), 6.94 (t, *J* = 8.6 Hz, 2H), 6.31 (s, 1H), 4.03–3.86 (m, 2H), 1.30 (d, *J* = 6.9 Hz, 12H); ^13^C NMR (101 MHz, Chloroform-*d*) δ 158.6 (d, *J_C-F_* = 241.2 Hz), 154.8, 135.4 (d, *J_C-F_* = 2.9 Hz), 121.8 (d, *J_C-F_* = 7.9 Hz), 115.2 (d, *J_C-F_* = 22.1 Hz), 45.6, 21.4; ^19^F NMR (376 MHz, Chloroform-*d*) δ -120.90; HRMS (ESI-TOF) *m/z*: [M + H]^+^ calcd for C_13_H_20_FN_2_O, 239.1554; found, 239.1567;

3-(4-chlorophenyl)-1,1-diisopropylurea (**5g**):

White solid (49.8 mg, 98%). ^1^H NMR (400 MHz, Chloroform-*d*) δ 7.32 (d, *J* = 8.9 Hz, 2H), 7.22 (d, *J* = 8.8 Hz, 2H), 6.29 (s, 1H), 4.03–3.90 (m, 2H), 1.32 (d, *J* = 6.9 Hz, 12H); ^13^C NMR (101 MHz, Chloroform-*d*) δ 154.4, 138.0, 128.7, 127.4, 121.0, 45.6, 21.5;

1,1-diisopropyl-3-(3-methoxyphenyl)urea (**5h**):

White solid (43.6 mg, 87%). ^1^H NMR (400 MHz, Chloroform-*d*) δ 7.20–7.14 (m, 2H), 6.85 (dd, *J* = 8.0, 1.2 Hz, 1H), 6.57 (dd, *J* = 8.2, 1.7 Hz, 1H), 6.27 (s, 1H), 4.05–3.95 (m, 2H), 3.81 (s, 3H), 1.33 (d, *J* = 6.9 Hz, 12H); ^13^C NMR (101 MHz, Chloroform-*d*) δ 160.2, 154.5, 140.7, 129.4, 111.7, 108.6, 105.1, 55.3, 45.4, 21.5;

1,1-diisopropyl-3-(3-(trifluoromethyl)phenyl)urea (**5i**):

White solid (21.3 mg, 37%), mp 151.2–152.7 °C. ^1^H NMR (400 MHz, Chloroform-*d*) δ 7.67 (s, 1H), 7.62–7.55 (m, 1H), 7.39 (t, *J* = 7.9 Hz, 1H), 7.29–7.25 (m, 1H), 6.37 (s, 1H), 4.07–3.94 (m, 2H), 1.36 (d, *J* = 6.9 Hz, 12H); ^13^C NMR (101 MHz, Chloroform-*d*) δ 154.2, 139.9, 131.2 (q, *J_C-F_* = 271.6 Hz), 129.3, 122.7, 119.1 (q, *J_C-F_* = 3.7 Hz), 116.1 (q, *J_C-F_* = 4.0 Hz), 45.7, 21.5; ^19^F NMR (376 MHz, Chloroform-*d*) δ -62.64. HRMS (ESI-TOF) *m/z*: [M + H]^+^ calcd for C_14_H_20_F_3_N_2_O, 289.1522; found, 289.1536;

3-(3-fluorophenyl)-1,1-diisopropylurea (**5j**):

White solid (23.9 mg, 50%), mp 118.5–119.8 °C. ^1^H NMR (400 MHz, Chloroform-*d*) δ 7.40–7.32 (m, 1H), 7.23–7.14 (m, 1H), 7.00 (dd, *J* = 8.3, 2.1 Hz, 1H), 6.72–6.64 (m, 1H), 6.39 (s, 1H), 4.03–3.89 (m, 2H), 1.32 (d, *J* = 6.6 Hz, 12H); ^13^C NMR (101 MHz, Chloroform-*d*) δ 163.2 (d, *J_C-F_* = 243.5 Hz), 154.2, 141.1 (d, *J_C-F_* = 11.2 Hz), 129.7 (d, *J_C-F_* = 9.6 Hz), 114.7 (d, *J_C-F_* = 2.9 Hz), 109.0 (d, *J_C-F_* = 21.3 Hz), 106.9 (d, *J_C-F_* = 26.4 Hz), 45.6, 21.5; ^19^F NMR (376 MHz, Chloroform-*d*) δ -112.35. HRMS (ESI-TOF) *m/z*: [M + H]^+^ calcd for C_13_H_20_FN_2_O, 239.1554; found, 239.1564;

1,1-diisopropyl-3-(*o*-tolyl)urea (**5k**):

White solid (43.0 mg, 92%), mp 136.8–138.7 °C. ^1^H NMR (400 MHz, Chloroform-*d*) δ 7.77 (dd, *J* = 8.1, 1.2 Hz, 1H), 7.23–7.14 (m, 2H), 7.04–6.96 (m, 1H), 6.07 (s, 1H), 4.11–3.99 (m, 2H), 2.28 (s, 3H), 1.36 (d, *J* = 6.9 Hz, 12H); ^13^C NMR (101 MHz, Chloroform-*d*) δ 154.8, 137.6, 130.3, 127.6, 126.7, 123.2, 122.3, 45.4, 21.5, 18.3. HRMS (ESI-TOF) *m/z*: [M + H]^+^ calcd for C_14_H_23_N_2_O, 235.1805; found, 235.1816;

3-(2-fluorophenyl)-1,1-diisopropylurea (**5l**):

White solid (40.0 mg, 84%), mp 109.1–110.7 °C. ^1^H NMR (400 MHz, Chloroform-*d*) δ 8.24–8.16 (m, 1H), 7.13–7.02 (m, 2H), 6.99–6.90 (m, 1H), 6.58 (s, 1H), 4.12–4.03 (m, 2H), 1.35 (d, *J* = 6.9 Hz, 12H); ^13^C NMR (101 MHz, Chloroform-*d*) δ 154.1, 152.3 (d, *J_C-F_* = 239.8 Hz), 128.0 (d, *J_C-F_* = 9.5 Hz), 124.5 (d, *J_C-F_* = 3.6 Hz), 122.0 (d, *J_C-F_* = 7.5 Hz), 121.1, 114.3 (d, *J_C-F_* = 19.2 Hz), 45.3, 21.4; ^19^F NMR (376 MHz, Chloroform-*d*) δ -133.39. HRMS (ESI-TOF) *m/z*: [M + H]^+^ calcd for C_13_H_20_FN_2_O, 239.1554; found, 239.1564;

3-(2-chlorophenyl)-1,1-diisopropylurea (**5m**):

White solid (49.6 mg, 95%). ^1^H NMR (400 MHz, Chloroform-*d*) δ 7.41–7.37 (m, 2H), 7.29 (d, *J* = 8.6 Hz, 1H), 7.06–6.98 (m, 1H), 6.27 (s, 1H), 4.05–3.95 (m, 2H), 1.34 (d, *J* = 6.9 Hz, 12H); ^13^C NMR (101 MHz, Chloroform-*d*) δ 154.6, 139.4, 128.8, 122.6, 119.7, 45.5, 21.5;

1,1-diethyl-3-phenylurea (**5n**):

White solid (36.8 mg, 96%). ^1^H NMR (400 MHz, Chloroform-*d*) δ 7.43–7.39 (m, 2H), 7.31–7.25 (m, 2H), 7.02 (t, *J* = 7.3 Hz, 1H), 6.40 (s, 1H), 3.41–3.35 (m, 4H), 1.22 (t, *J* = 7.1 Hz, 6H); ^13^C NMR (101 MHz, Chloroform-*d*) δ 154.7, 139.4, 128.8, 122.8, 119.9, 41.6, 13.9;

3-phenyl-1,1-dipropylurea (**5o**):

White solid (35 mg, 80%). ^1^H NMR (400 MHz, Chloroform-*d*) δ 7.43–7.37 (m, 2H), 7.31–7.25 (m, 2H), 7.06–6.97 (m, 1H), 6.41 (s, 1H), 3.31–3.24 (m, 4H), 1.71–1.61 (m, 4H), 0.96 (t, *J* = 7.4 Hz, 6H); ^13^C NMR (101 MHz, Chloroform-*d*) δ 155.0, 139.4, 128.8, 122.7, 119.8, 49.4, 21.9, 11.4;

1,1-dibutyl-3-phenylurea (**5p**):

White solid (41.6 mg, 84%). ^1^H NMR (400 MHz, Chloroform-*d*) δ 7.44–7.38 (m, 2H), 7.30–7.25 (m, 2H), 7.05–6.99 (m, 1H), 6.38 (s, 1H), 3.34–3.28 (m, 4H), 1.65–1.58 (m, 4H), 1.43–1.34 (m, 4H), 0.98 (t, *J* = 7.3 Hz, 6H); ^13^C NMR (101 MHz, Chloroform-*d*) δ 155.0, 139.4, 128.8, 122.7, 119.7, 47.5, 30.8, 20.2, 13.9;

1-ethyl-3-phenyl-1-propylurea (**5q**):

Colorless liquid (34.0 mg, 83%), mp 56.3–57.6 °C. ^1^H NMR (400 MHz, Chloroform-*d*) δ 7.43–7.37 (m, 2H), 7.31–7.25 (m, 2H), 7.02 (t, *J* = 7.4 Hz, 1H), 6.39 (s, 1H), 3.42–3.34 (m, 2H), 3.30–3.24 (m, 2H), 1.71–1.61 (m, 2H), 1.22 (t, *J* = 7.1 Hz, 3H), 0.96 (t, *J* = 7.4 Hz, 3H); ^13^C NMR (101 MHz, Chloroform-*d*) δ 154.9, 139.4, 128.8, 122.7, 119.8, 48.8, 42.1, 22.0, 13.8, 11.4; HRMS (ESI-TOF) *m/z*: [M + H]^+^ calcd for C_12_H_19_N_2_O, 207.1492; found, 207.1499;

1-cyclohexyl-1-ethyl-3-phenylurea (**5r**):

White solid (44.0 mg, 89%), mp 123.5–124.7 °C. ^1^H NMR (400 MHz, Chloroform-*d*) δ 7.44–7.39 (m, 2H), 7.31–7.25 (m, 2H), 7.01 (t, *J* = 7.3 Hz, 1H), 6.41 (s, 1H), 4.14–4.03 (m, 1H), 3.33–3.25 (m, 2H), 1.85–1.76 (m, 4H), 1.72–1.64 (m, 1H), 1.47–1.33 (m, 4H), 1.25 (t, *J* = 7.2 Hz, 3H), 1.18–1.06 (m, 1H); ^13^C NMR (101 MHz, Chloroform-*d*) δ 154.8, 139.5, 128.8, 122.7, 119.9, 54.8, 36.9, 31.5, 26.0, 25.6, 16.1; HRMS (ESI-TOF) *m/z*: [M + H]^+^ calcd for C_15_H_23_N_2_O, 247.1805; found, 247.1815;

1,1-dicyclohexyl-3-phenylurea (**5s**):

White solid (54.0 mg, 90%). ^1^H NMR (400 MHz, Chloroform-*d*) δ 7.40–7.36 (m, 2H), 7.31–7.25 (m, 2H), 7.04–6.98 (m, 1H), 6.32 (s, 1H), 3.55–3.45 (m, 2H), 1.89–1.81 (m, 6H), 1.80–1.75 (m, 6H), 1.69 (d, *J* = 13.0 Hz, 2H), 1.42–1.31 (m, 4H), 1.22–1.11 (m, 2H); ^13^C NMR (101 MHz, Chloroform-*d*) δ 154.9, 139.4, 128.8, 122.5, 119.7, 55.5, 31.9, 26.4, 25.6;

*N*-phenyl-3,4-dihydroisoquinoline-2(1H)-carboxamide (**5t**):

White solid (45.1 mg, 89%). ^1^H NMR (400 MHz, Chloroform-*d*) δ 7.46–7.42 (m, 2H), 7.32–7.27 (m, 2H), 7.25–7.17 (m, 3H), 7.15–7.10 (m, 1H), 7.05 (t, *J* = 7.4 Hz, 1H), 6.75 (s, 1H), 4.67 (s, 2H), 3.72 (t, *J* = 5.9 Hz, 2H), 2.91 (t, *J* = 5.9 Hz, 2H); ^13^C NMR (101 MHz, Chloroform-*d*) δ 155.2, 139.2, 135.0, 133.3, 128.9, 128.4, 126.8, 126.5, 126.4, 123.1, 120.3, 45.8, 41.6, 29.0;

1-benzyl-1-ethyl-3-phenylurea (**5u**):

Colorless liquid (42.8 mg, 84%). ^1^H NMR (400 MHz, Chloroform-*d*) δ 7.42–7.38 (m, 2H), 7.37–7.30 (m, 5H), 7.29–7.24 (m, 2H), 7.06–7.00 (m, 1H), 6.40 (s, 1H), 4.59 (s, 2H), 3.52–3.45 (m, 2H), 1.24 (t, *J* = 7.1 Hz, 3H); ^13^C NMR (101 MHz, Chloroform-*d*) δ 155.4, 139.2, 137.7, 129.0, 128.8, 127.7, 127.1, 122.9, 119.9, 50.3, 42.5, 13.5. HRMS (ESI-TOF) *m/z*: [M + H]^+^ calcd for C_16_H_19_N_2_O, 255.1492; found, 255.1500;

1-benzyl-1-isopropyl-3-phenylurea (**5v**):

White solid (48.8 mg, 91%), mp 108.8–109.9 °C. ^1^H NMR (400 MHz, Chloroform-*d*) δ 7.45–7.37 (m, 4H), 7.37–7.32 (m, 1H), 7.25–7.19 (m, 4H), 7.02–6.96 (m, 1H), 6.36 (s, 1H), 4.86–4.73 (m, 1H), 4.47 (s, 2H), 1.24 (d, *J* = 6.8 Hz, 6H); ^13^C NMR (101 MHz, Chloroform-*d*) δ 155.8, 139.3, 138.2, 129.2, 128.7, 127.8, 126.4, 122.8, 119.8, 46.4, 45.5, 20.8. HRMS (ESI-TOF) *m/z*: [M + H]^+^ calcd for C_17_H_21_N_2_O, 269.1648; found, 269.1657;

1,3-diphenylurea (**5w**):

White solid (22.5 mg, 52%). ^1^H NMR (400 MHz, DMSO-*d*_6_) δ 8.7 (s, 2H), 7.5 (dd, *J* = 8.6, 1.2 Hz, 4H), 7.3–7.2 (m, 4H), 7.0–6.9 (m, 2H); ^13^C NMR (101 MHz, DMSO-*d*_6_) δ 153.0, 140.2, 129.2, 122.3, 118.6.

## 4. Conclusions

In conclusion, we have disclosed a visible-light-promoted external catalyst-free procedure for the decarboxylation of dioxazolones to synthesize various phosphinimidic amides and ureas. The method has the advantages of no additional transition metals, economic raw materials, mild reaction conditions, and easy operation. It could be reasoned that the ppm Fe in the reaction mixture played a significant role in this transformation.

## Data Availability

The data presented in this study are available on request from the corresponding author.

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
