# Peer review of "Visible-Light-Induced Decarboxylation of Dioxazolones to Phosphinimidic Amides and Ureas"

_molecules, 2022, doi:10.3390/molecules27123648_

Round 1
Reviewer 1 Report
This paper is devoted to the light induced transformation of dioxazolones into phosphazenes, ureas and amides. Discovered reactions are easy to carry and provide good to excellent yields of products. Obtained products are valuable chemicals, especially asymmetric ureas. The possibility of C-N bond construction is of special importance, since the introduction of amino groups in the organic substrates under mild conditions is very much requested by pharmaceutical industry and is rather challenging chemistry. Altogether it is excellent research which would be of great interest for the readers of “Molecules”. However, there are several minor issues to be addressed:
- Among all rather similar yields compound 5i was obtained with surprisingly low yield. Is there any explanation for this case?
- The idea with iron catalysis sounds reasonable, since technically everything is contaminated with iron (good example is oxidation of benzaldehyde on air). However, this feature should be experimentally proven. Did authors try to add additional iron catalyst to check whether is facilitates the reaction?
- Since molecules has no limitation for the paper size, it would be better to include spectral data of products to the manuscript for the convenience of readers.
Based on the above mentioned I recommend to accept this paper for publication after minor revision.
Reviewer 2 Report
This manuscript describes the synthesis of phosphinimidic amides and ureas via decarboxylation of dioxazolones under catalyst-free conditions. Yu and Bao reported the same chemistry in 2021. However, these results are interesting and studied carefully with good substrate scope for phosphinimidic amides and ureas with good yields. All the new compounds are fully characterized and NMRS are pure. This manuscript can be published in Molecules. To improve the quality of the manuscript authors, need to address the below comments.
1). It seems the authors did ICP-MS experiment to find the Fe content only for the synthesis of the phosphinimidic amide. What about the ureas synthesis?
2). It would be clearer if the authors had done the same experiment for starting materials dioxazolone, PPh3 and amine instead of after reaction.
3). DCM must be CH2Cl2
4). Authors should provide all NMRS with a uniform scale.
5). Several space issues in the characterization data.
6). There are some errors in supporting information Schemes and figure numbering. Both supporting information files must have a continuation number.
7). Figure 1. sensitivity assessment must be Figure S1.
Reviewer 3 Report
This paper describes the photochemical transformation of 1,4,2-dioxazol-5-ones with tertiary phosphines and secondary amines to construct N=P and N–C bonds.The formation N-acyl nitrene speciesfrom dioxazolonesis a well-known process in which transition metals promote decarboxylation. The authors underscore that the “external catalyst-free” conditions are new findings, as denoted in the manuscript title; however, the mechanistic conclusion is the catalysis with contaminated iron. In my view, it would be an overstatement, and I recommend removing the term “external catalyst-free” from the title.
Considering the significance of photocatalytic protocol without precious metals and organic dye additives,this manuscript would be of interest to the readers and merits publication in Molecules, after addressing the following points:
(1) In Tables 2 and 3; revise the table captions to be more specific
(2) Regarding the urea synthesis in Table 3, it is advisable to compare with the reported results in ref. 32. How about the primary amines and aniline derivatives? Further experiments to expand the scope and discussion are highly recommended.
(3) Line 98; a similar reaction to Scheme 2a was performed in ref. 29. It needs to be mentioned and cited.
(4) Cite the following paper describing the mild thermally induced decarboxylation together with a hydrolytic process of dioxazolones leading to ureas: V. Bizet, L. Buglioni, C. Bolm, Angew. Chem. Int. Ed. 2014, 53, 5639.
Round 2
Reviewer 2 Report
In the updated manuscript authors addressed all the reviewer comments. However, It is surprising to have Iron traces in all the starting materials.
I request authors provide the commercial sources of all starting materials, including those required for the synthesis of diaoxazolone in the manuscript.
While submitting the manuscript, authors could give expert comments on the possibility of Iron contamination in all the starting materials and starting materials for all substrates.
Reviewer 3 Report
Theresubmitted paper has been adequately revised based on the reviewers’ comments and the authors added some more citations for further refinement. Overall, I’m happy to support publication in Molecules.
Author Response
Thank you